# Impact of Runoff Schemes on Global Flow Discharge: A Comprehensive Analysis Using the Noah-MP and CaMa-Flood Models

Mohamed Hamitouche[1,2,3], Giorgia Fosser[1], Alessandro Anav [2,3], Cenlin He[4], Tzu-Shun Lin [4]

[1]University School for Advanced Studies IUSS, Pavia, Italy
[2]Climate Modeling Laboratory ENEA – Italian National Agency for New Technologies, Energy and Sustainable Economic Development. CR Casaccia, Viale Anguillarese 301, 00123, Santa Maria di Galeria (Rome), Italy
[3]ICSC Italian Research Center on High-Performance Computing, Big Data and Quantum Computing, Bologna, Italy
[4]Research Applications Laboratory, NSF National Center for Atmospheric Research, Boulder, Colorado, USA

*Correspondence to*: Mohamed Hamitouche (mohamed.hamitouche@iusspavia.it)

**Abstract.** Accurate estimation of flow discharge is crucial for hydrological modelling, water resources planning, and flood prediction. This study examines seven common runoff schemes within the widely-used Noah-MP land surface model and evaluates their performance, using ERA5-Land runoff data as a benchmark for assessing runoff and in-situ streamflow observations for evaluating discharge across the globe. Then, to assess the sensitivity of global river discharge to runoff, we simulate the discharge, using the CaMa-Flood model, across various climatic regions. The results indicated significant variability in the accuracy of the runoff schemes, with model experiments that use TOPMODEL-based runoff schemes, which are based on topography, underestimating runoff across many regions, particularly in the Northern Hemisphere, while experiments using the other runoff schemes including default Schaake free drainage scheme from Noah, BATS (Biosphere-Atmosphere Transfer Scheme), Variable Infiltration Capacity (VIC) scheme, and Xinanjiang scheme showed improved performance. Dynamic VIC consistently overestimated runoff globally. Seasonal analysis reveals substantial regional and seasonal variability. ERA5-Land and several Noah-MP schemes successfully replicated general discharge patterns of in-situ observations, with ERA5-Land and Noah-MP Schaake-scheme simulations closely aligning with observed data. The Noah-MP simulations demonstrated robust versatility across various land covers, soil types, basin sizes, and topographies, indicating its broad applicability. Despite overall good performance, significant biases in high-flow extremes highlight the need for continued model improvement or calibration. These findings are critical for improving global hydrological models, which are essential for developing more reliable water resource management strategies and adapting to the growing challenges posed by climate change, such as shifts in water availability and extreme flood events.

## 1 Introduction

Accurate estimation of flow discharge, a fundamental component of global hydrological cycle and a critical water flux from land to ocean (Stephens et al., 2020), is crucial for effective hydrological modelling, water resources planning, flood prediction, and sustainable water management practices. It is essential for flood prediction, aiding in alerts, evacuation plans, and response strategies (Nguyen et al., 2022). Water supply planning relies on it for equitable allocation across domestic, agricultural, and industrial needs, ensuring sustainability. Ecosystem health benefits from maintaining proper water levels in habitats,

safeguarding biodiversity and stability. Infrastructure design and maintenance use accurate estimates for resilient structures against varying conditions. Hydropower generation optimisation and cost management depend on it. Water quality management relies on precise estimation to guide monitoring and pollution control (Zhang et al., 2011). In the context of climate change, it is essential for understanding shifting hydrological patterns and for adapting strategies. Therefore, policies and regulations centred on discharge data necessitate accurate estimation for compliance, equitable allocation, and resource

distribution.

On the other hand, runoff and groundwater dynamics are among the most influential physical processes for land surface hydrological simulations, as demonstrated by various on-site and regional simulations (Gan et al., 2019; Li et al., 2020; Zhang et al., 2016, 2021a, b; Zheng et al., 2019). Past research has acknowledged the intricate interconnection between runoff and flow discharge, highlighting the propagation of uncertainty from runoff to discharge (David et al., 2019). Recognising their

pivotal role, runoff schemes serve as vital components of land surface and hydrological models (Sheng et al., 2017). These models represent the processes governing the conversion of precipitation into runoff. This understanding underscores the significant influence that the selection of a particular runoff scheme can have on flow discharge patterns, yielding profound implications for water resources availability and management. Consequently, there is a pressing demand to conduct a comprehensive quantification of the diverse impacts that various runoff schemes can have on flow discharge.

Runoff schemes, each grounded in distinct hydrological theories and assumptions, can exert diverse influences on flow discharge dynamics. These effects can stem from variations in representation of hydrological processes such as surface runoff, subsurface flow, infiltration, root water uptake, groundwater dynamics, and stream-aquifer interactions (Clark et al., 2015). For instance, schemes that emphasise surface runoff may lead to altered flow pathways and the timing of peak flows, impacting downstream water availability. Conversely, schemes incorporating subsurface processes may enhance groundwater recharge,

potentially modifying baseflow contributions and seasonal streamflow patterns.

Moreover, the intricate interactions between runoff schemes, climatic conditions, land cover types, and soil properties, further accentuate their potential impacts on flow discharge (Zipper et al., 2018). Different runoff schemes may exhibit varying sensitivities to climatic variability, resulting in disparate responses to changing precipitation patterns, temperature shifts, and extreme events. Two mechanisms, saturation excess and infiltration excess, jointly contribute to the generation of runoff (Yang

et al., 2015). Runoff schemes based on the saturated excess assumption are valid in humid and pervious areas; however, this assumption has limitations in dry and impervious areas where overland flow dominates due to excess infiltration (Ren-Jun,

1992). This underscores the importance of scrutinising runoff scheme behaviour under diverse climate conditions to unravel the complexities of their effects on flow discharge.

Despite prior research has contributed valuable insights into runoff scheme impacts on flow discharge, limitations still persist.
Many studies have focused on individual runoff schemes (e.g.: Li et al., 2022) and confined their investigations to specific hydroclimatic contexts (e.g.: Hagemann and Stacke, 2023; Liang et al., 2019) or catchments (e.g.: Rummler et al., 2022; Zheng et al., 2017) that overlook potential interactions and synergies of the diverse runoff schemes when operating globally. A singular focus can limit insights into how runoff schemes collectively shape flow discharge dynamics. Additionally, the generalisation and extrapolation of findings to broader global contexts can be challenging due to geographical and climatic
variability, differences in hydrological regimes, land cover, soil types, and other factors.

Given the above-mentioned limitations, this paper aims to elucidate, at global scale, the impact of different runoff schemes on river discharge estimation. To that purpose, we firstly evaluate the performances of seven distinct runoff schemes within the Noah-MP Land Surface Model (LSM), then we simulate flow discharge with the CaMa-Flood River routing model to assess how different runoff schemes affect flow discharge magnitude and dynamics. Our study transcends the boundaries of
individual schemes and specific regions, highlighting the need for a holistic assessment that contributes to improved hydrological modelling and management practices.

## 2 Materials and methods

### 2.1 Noah-MP Land Surface Model

In this study, the Noah-MP LSM was applied to simulate global-scale runoff. The Noah-MP LSM is a spatially distributed 1-
D model specifically designed to address the vertical routing of surface and subsurface water flow in response to atmospheric forcing, all within individual grid cells. This versatile model incorporates four soil layers, extending to a maximum depth of 2 metres, each with default thicknesses of 0.1 metres, 0.3 metres, 0.6 metres, and 1 metre. It solves Richard's equation to compute the dynamics of soil water content (Chen et al., 1996). In addition to soil water dynamics, Noah-MP computes various surface energy flux components, accounts for gravitational drainage at the lowest soil layer, and handles the partitioning of surface
water into infiltration and surface runoff. These computations are facilitated through a range of parameterisation approaches (for detailed information, refer to (He et al., 2023b)).

Noah-MP is designed to operate in both uncoupled and coupled modes, seamlessly integrating with atmospheric and/or hydrological models at sub-daily time scales and high spatial resolutions, including point-scale, regional and global simulations. This versatility enables its use in a variety of hydrological, weather, and climate models, offering adaptability
across a wide range of spatial and temporal scales while ensuring proper integration in both time and space (He et al., 2023a). Moreover, Noah-MP offers a multi-parameterisation framework that encompasses over 4608 combinations of more than 10 physical processes (Niu et al., 2011) that govern interactions at the land-atmosphere interface. These processes include modules for vegetation dynamics, soil moisture, snowpack accumulation and melt, energy balance, and more. The incorporation of

multiple physics-based processes within Noah-MP allows for a comprehensive representation of real-world conditions and
facilitates ensemble experiments with the multi-physics model for uncertainty assessment and testing competing hypotheses
(Li et al., 2020; Zhang et al., 2016).

### 2.1.1 Noah-MP Parametrisation and Runoff Schemes Overview

The community Noah-Multi parameterisation Land Surface Model (Noah-MP LSM) (Niu et al., 2011; Yang et al., 2011)
evolved from the Noah LSM (Chen et al., 1996, 1997; Chen and Dudhia, 2001; Ek et al., 2003), incorporating enhanced
physical representations and treatments for dynamic vegetation, canopy interactions, radiative transfer, multi-layer snowpack
physics, and soil-hydrological processes. In this study, the last modernised version of Noah-MP (v5.0) with enhanced
modularity, interoperability and applicability (He et al., 2023a) was utilised.

As our main focus is on runoff, this study was conducted using the default parametrisation scheme combination outlined in
the Noah-MP v5.0 public release code, in uncoupled mode. The main default options include: the Noah-type (Ek et al., 2003)
for soil moisture factor, stomatal resistance and evapotranspiration; Monin–Obukhov (M–O) Similarity Theory (MOST)
(Monin and Obukhov, 1954) for surface layer exchange coefficient, canopy gaps calculated from the vegetated fraction (gap
= 1-VegFrac) (Dickinson, 1983; Sellers, 1985) and for canopy radiation transfer; the Ball–Berry scheme (Ball et al., 1987;
Bonan, 1996) for canopy stomate resistance; Sakaguchi and Zeng's scheme (Sakaguchi and Zeng, 2009) for ground resistance
to evaporation/sublimation; the Canadian land surface scheme (CLASS) type (Verseghy, 1991) for ground snow surface
albedo, hydraulic properties calculated from total soil water and ice (Niu and Yang, 2006) for frozen soil permeability; the
general form of the freezing-point depression equation (Niu and Yang, 2006) for soil supercooled liquid water; and the dynamic
vegetation model (Dickinson et al., 1998) turned off but using maximum vegetation fraction and a look-up table for leaf area
index.

In this study, we conducted seven simulations with different runoff and groundwater schemes. Each experiment (henceforth
EXP) was numbered according to the Noah-MP runoff options as follows:

1) TOPMODEL with groundwater (Niu et al., 2007),

2) TOPMODEL with an equilibrium water table (Niu et al., 2005),

3) Original Noah free drainage or Schaake's runoff (Schaake et al., 1996),

4) BATS surface and subsurface runoff (Yang and Dickinson, 1996),

6) Variable Infiltration Capacity (VIC) Model surface runoff scheme (Liang et al., 1994),

7) Xinanjiang Infiltration and surface runoff scheme (XAJ, (Jayawardena and Zhou, 2000)),

8) Dynamic VIC surface runoff scheme (Liang and Xie, 2003).

The Miguez-Macho&Fan groundwater scheme ((Fan et al., 2007; Miguez-Macho et al., 2007); Noah-MP runoff option 5) was
not included in this analysis due to unavailable global riverbed data and other essential inputs.

Infiltration excess and saturation excess runoff generation processes are the key factors leading to the difference among the selected options. This distinction has a direct impact on the velocity of surface runoff and bottom drainage fluxes, leading to the removal of water mass and shifts in liquid soil water content (Chang et al., 2020).

The TOPMODEL with groundwater approach (EXP1) utilises a simplified groundwater modelling method outlined by (Niu et al., 2007). In this method, vertical recharge to an unconfined aquifer is estimated through a parameterisation of Darcy's Law. Groundwater storage calculations are then employed to determine the grid-scale water table depth (dwt), which is subsequently converted into the saturated surface fractional area (fsat), given as:

$$fsat = fsat_{max} \times e^{-0.5 \times F_{decay} \times dwt} \qquad (1)$$

where $F_{decay}$ is the runoff decay factor, and $fsat_{max}$ is the maximum saturated fraction of soil surface (assigned a fixed, unitless value of 0.38). Surface runoff is calculated using a saturation excess runoff generation process, where fsat is multiplied by the precipitation that falls on the soil surface. Subsurface runoff is assumed to be proportional to $exp[-F_{decay}(dwt)]$.

TOPMODEL with equilibrium runoff (EXP2) calculations are similar to the previous scheme, with the key difference being that dwt is determined using an equilibrium water table calculation rather than a dynamic groundwater balance (Niu et al., 2005, 2011).

Unlike TOPMODEL-based schemes, the Schaake and BATS parameterisations do not account for water table dynamics, but use a gravitational free-drainage baseflow approach as a bottom boundary condition. These two approaches differ in their treatment of surface runoff. The Schaake approach (EXP3) employs the infiltration excess surface runoff method described by (Schaake et al., 1996), which is based on an adaptation of the Soil Conservation Service curve number method. In this approach, the surface runoff prediction is notably sensitive to the Noah-MP parameter REFKDT (Niu and Yang, 2011), which controls the influence of pre-storm surface soil moisture conditions and is linearly related to the Kdt parameter described by (Schaake et al., 1996). The BATS physics scheme (EXP4), following (Yang and Dickinson, 1996), parameterises the fraction of incident precipitation converted into runoff as the fourth power of the degree of saturation in the top 2 metres of the soil column. The gravitational drainage is parametrised in Schaake as the product of the soil drainage slope index ($S_{drain}$) and the soil hydraulic conductivity (DK), while in BATS as $(1 - f_{imp,max})DK$ where $f_{imp,max}$ is the maximum soil impermeability fraction throughout the soil column.

The VIC scheme (EXP6) calculates the saturation excess surface runoff in surface soil layers (two first layers) based on a variable infiltration capacity (i) function given by:

$$i = i_{max} \times (1 - (1 - A)^{1/b}) \qquad (2)$$

where $i_{max}$ is the maximum infiltration capacity, A is the fraction of saturated soil in a grid, and b is a curve shape parameter. The surface runoff ($R_s$) estimated by VIC runoff scheme is given following Eq. (12), Eq. (13) and Eq. (14) in Table 1.

The XAJ infiltration and surface runoff scheme (EXP7) introduces a distinctive approach to hydrological modelling. It addresses the saturation excess runoff generation by incorporating the concept of variable contributing area and using a double parabolic curve to represent the spatial distribution of tension water capacity (maximum soil water deficit, i.e., the difference between field capacity and wilting point), which is considered the essence of the XAJ model (Fang et al., 2017). In the selected Noah-MP version, the runoff generation process within the Xinanjiang scheme acts on the two first soil layers, resulting in the separation of runoff into two components: surface and subsurface runoff, distinguishing between impervious and pervious surfaces (He et al., 2023b). Surface runoff generated from impervious area ($R_{im}$) is determined by the product of the fraction of imperviousness due to frozen soil ($A_{im}$) and the effective precipitation ($P_e$) (i.e., mean water input on the soil surface). Surface runoff from permeable soil ($R_p$) is given by Eq. (3):

$$R_p = R \times \left( 1 - \left( 1 - \frac{S}{S_{max}} \right)^{E_x} \right) \tag{3}$$

where R is runoff filled from tension water areas, and expressed as:

$$R = \begin{cases} P_i \times \left[ (0.5 - a)^{1-b} \times \left( \frac{W}{W_{max}} \right)^b \right], if\ 0 \leq \frac{W}{W_{max}} \leq 0.5 - a \\ P_i \times \left[ 1 - (0.5 + a)^{1-b} \times \left( 1 - \frac{W}{W_{max}} \right)^b \right], 0.5 - a < \frac{W}{W_{max}} \leq 1 \end{cases} \tag{4}$$

with $P_i$ being the fraction of effective precipitation falling on pervious area ($1-A_{im}$). W and $W_{max}$ are respectively the current and maximum tension water storage. S and $S_{max}$ are respectively the current and maximum free water storage. $E_x$, a and b are shape parameters.

(Liang and Xie, 2001, 2003) extended the VIC model to include the infiltration excess runoff generation process (EXP8). The new parametrisation (Dynamic VIC) dynamically represents both Hortonian and Dunnian runoff generation processes by considering effects of sub-grid spatial heterogeneity of soil properties. The saturation excess runoff ($R_{se}$) is calculated for the saturated area fraction ($A_s$) following the concept used in VIC, while the infiltration excess runoff ($R_{ie}$) is computed for $1-A_s$ area fraction following Eq. (16) in Table 1.

The subsurface runoff in VIC, Dynamic VIC and XAJ models is drainage-dependent, and calculated as per Schaake scheme. The seven experiments, the runoff scheme used for each experiment and their corresponding surface and subsurface runoff equations are summarised in Table 1.

**Table 1: Summary of the experiments (EXPs), runoff schemes, and corresponding equations**

| EXP | Runoff Scheme | Surface Runoff ($R_s$) Equation | Subsurface Runoff ($R_{sub}$) Equation |
|---|---|---|---|
| 1 | TOPMODEL with groundwater | $$R_s = P_e \times \left[\left(1 - f_{imp}(1)\right) \times fsat + f_{imp}(1)\right] \quad (5)$$ | $$R_{sub} = (1 - f_{imp,max}) \times C_{baseflow} \times e^{-I_{topo}} \times e^{(-F_{decay} \times dwt)} \quad (6)$$ |
| 2 | TOPMODEL with an equilibrium water table | Equation (5) | Equation (6) |
| 3 | Schaake | $$R_s = P_e - Q_{infil,max} \quad (7)$$ $$R_s = P_e \times \left[1 - \frac{w_{soil,tot} \times (1 - e^{-Kdt \times \Delta t})}{P_e \times \Delta t + w_{soil,tot} \times (1 - e^{-Kdt \times \Delta t})}\right] \quad (8)$$ The $Q_{infil,max}$ is further corrected for frozen soil as follows: $$Q_{infil,max} = min(max(Q_{infil,max} \times f_{imp}; DK); P_e) \quad (9)$$ | $$R_{sub} = S_{drain} \times DK \quad (10)$$ |
| 4 | BATS | Equation (5) | $$R_{sub} = (1 - f_{imp,max}) \times DK \quad (11)$$ |
| 6 | Variable Infiltration Capacity (VIC) | $$\text{If } i + P_e \geq i_{max}: R_s = P_e - W_{max} + W \quad (12)$$ $$\text{If } i + P_e \leq i_{max}:$$ $$R_s = P_e - W_{max} + W + W_{max} \times \left[1 - \frac{i + P_e}{i_{max}}\right]^{(1+b)} \quad (13)$$ $$\text{If } i_{max} = 0: R_s = P_e \quad (14)$$ | Equation (10) |
| 7 | Xinanjiang (XAJ) | $$R_s = (P_e \times A_{im}) + R \times \left(1 - \left(1 - \frac{S}{S_{max}}\right)^{E_x}\right) \quad (15)$$ | Equation (10) |
| 8 | Dynamic VIC | $R_s = R_{ie} + R_{se}$ <br> With: | Equation (10) |

$$R_{ie} = \begin{cases} if \ \dfrac{P - R_{se}}{f_m \times \Delta t} \leq 1, \\[2mm] P - R_{se} - f_{mm} \times \Delta t \times \left[1 - \left(1 - \dfrac{P - R_{se}}{f_m \times \Delta t}\right)^{b+1}\right] \\[2mm] otherwise, \\[1mm] P - R_{se} - f_{mm} \times \Delta t \end{cases} \quad (16)$$

And:

$$R_{se} = \begin{cases} if \ 0 \leq y < i_m - i_0, \\[2mm] y - \dfrac{i_m}{b+1} \times \left[\left(1 - \dfrac{i_0}{i_m}\right)^{b+1} - \left(1 - \dfrac{i_0 + y}{i_m}\right)^{b+1}\right] \\[2mm] if \ i_m - i_0 \leq y < P, \\[1mm] R_{se}|_{y = i_m - i_0} + y - (i_m - i_0) \end{cases} \quad (17)$$

$f_{imp}(i)$: the $i^{th}$ soil layer impermeable fraction; $Q_{infil,max}$: the maximum soil infiltration rate; $w_{soil,tot}$: the sum of the maximum holdable soil water content in the unit of depth; Kdt: a coefficient for computing maximum soil infiltration rate; P: the amount of precipitation over a time step $\Delta t$; $f_{mm}$: the average potential infiltration rate over the 1-As area estimated based on the Philip infiltration scheme (Liang and Xie, 2003); $f_m$: the maximum potential infiltration rate; y: vertical depth; $i_0$: the point soil moisture capacity corresponding to the initial soil moisture; $i_m$: the maximum point soil moisture capacity; $C_{baseflow}$: a baseflow coefficient; $I_{topo}$: the gridcell mean topographic index.

### 2.1.2 Input data

ERA5-Land (Muñoz-Sabater et al., 2021), at 0.1-degree resolution, represents the inaugural operational land product in the European reanalysis (ERA) series. It was derived from high-resolution global numerical simulations conducted by the European Centre for Medium-Range Weather Forecasts (ECMWF). These simulations were driven by downscaled meteorological data sourced from the ERA5 climate reanalysis, which includes adjustments for elevation to enhance the accuracy of near-surface thermodynamic conditions. As a result, ERA5-Land offers a consistent depiction of water and energy cycles across the Earth's land surface (Li et al., 2022).

In this study, meteorological variables, including 10m wind speed, 2m air temperature, air humidity, surface pressure, longwave and shortwave downward radiation, and total precipitation, were extracted from the ERA5-Land hourly dataset. Subsequently, these variables were regridded to a spatial resolution of 0.2° to provide forcing data for the Noah-MP LSM. The soil water content simulations were performed over four distinct soil layers with depths corresponding to ECMWF model specifications: 0-7 cm, 7-28 cm, 28-100 cm, and 100-289 cm.

Land cover data was defined using the Modified International Geosphere-Biosphere Programme (IGBP) Moderate Resolution Imaging Spectroradiometer (MODIS) 20-category vegetation dataset, which covers the entire globe with 500m (15 seconds) grid intervals, while soil types were mainly determined by the State Soil Geographic (STATSGO)–FAO soil texture data (Miller and White, 1998).

ERA5-Land runoff was also used as a reference dataset for evaluating the Noah-MP simulated runoff. It is generated using the Tiled ECMWF Scheme for Surface Exchanges over Land incorporating land surface hydrology (H-TESSEL), and represents the total water volume accumulated over the forecast period, divided into surface and subsurface components (Liu et al., 2024). Surface runoff is generated when the maximum infiltration rate is exceeded, as described by the Arno scheme (Dümenil and Todini, 1992), while subsurface water fluxes are governed by Darcy's law, assuming free drainage at the bottom boundary

condition (Balsamo et al., 2009; Wipfler et al., 2011).

A recent study by (Dutta and Markonis, 2024) evaluated the performance of ERA5-Land runoff and found that it performs well in specific regions like Central Europe, India, and Southern North America, with a generally accurate representation of global runoff patterns. However, the study highlights significant inaccuracies in arid regions and similar climates. Additionally, ERA5-Land struggles with extreme events, often failing to generate high runoff following intense precipitation. Although with

these limitations that should be considered using ERA5-Land for terrestrial hydrological studies, ERA5-Land has the big advantage to provide atmosphere-coherent runoff at hourly resolution and high spatial resolution allowing for the validation of Noah-MP.

Prior to conducting runoff evaluation simulations, a 15-year spin-up of the Noah-MP LSM was performed for each model runoff experiment. This involved repeating the 1980-1984 interval three times to reach model stability and enhance accuracy

in simulating runoff and related hydrological processes. Following the spin-up phase, model simulations forced by hourly ERA5-Land data were carried out for the period 1985-2023.

## 2.2 CaMa-Flood River Routing Model

As Noah-MP LSM does not account for horizontal water exchanges, it necessitates supplementation with lateral flow algorithms to achieve accurate simulations of the river discharge. We used the Catchment-based Macro-scale Floodplain

model, CaMa-Flood (Yamazaki et al., 2011) due to its ability to simulate global temporal variations and discharge peaks. CaMa-Flood was proved to realistically simulate river water levels and the hydrodynamics of floodplain inundation (Hirabayashi et al., 2013).

Functioning as a global-scale distributed river model, CaMa-Flood relies on runoff input from a land surface model to simulate water storage and river discharge across a predefined river network map, available at different spatial resolutions: 0.25, 0.1,

0.0833, 0.05 and 0.0166 degree. The model discretises river basins into unit catchments, each delineated by sub-grid river and floodplain topography parameters, providing a nuanced representation of floodplain inundation at a sub-grid scale (Yamazaki et al., 2014). River discharge is calculated using the local inertial equation, accounting for the backwater effect and ensuring a more accurate portrayal of river dynamics (Bates et al., 2010; Yamazaki et al., 2013).

In its calculations, CaMa-Flood employs the Manning's friction coefficient for main river channels, set at 0.03. The model

dynamically adjusts the calculation time step to meet the Courant-Friedrichs-Lewy condition, ensuring computational stability (Bates et al., 2010; Yamazaki et al., 2013). Additionally, a channel bifurcation flow scheme enhances the model's capability to simulate intricate flow dynamics, particularly in mega deltas. This scheme automatically incorporates bifurcation channels

into the global river network map, extracting information from the HydroSHEDS flow direction map and the SRTM3 DEM (Yamazaki et al., 2014).

In this study, the CaMa-Flood model at 0.0833 degree spatial resolution was used. To ensure model stability, CaMa-Flood underwent a dedicated spin-up phase, synchronised with the third iteration of the Noah-MP spin-up, spanning a single 5-year interval from 1980 to 1984. Subsequently, daily runoff data simulated by the seven Noah-MP experiments were interpolated to match the resolution of the CaMa-Flood model, as in the spin-up phase, and used to drive river discharge simulations spanning from 1985 to 2023.

Both Noah-MP and CaMa-Flood models were set up over a global domain covering the land areas between 180°W to 180°E and 60°S to 90°N.

Additionally, ERA5-Land daily runoff was also used to simulate the daily river discharge. This assessment followed a similar procedure, including data interpolation, with a 5-year spin-up period from 1980 to 1984, after which CaMa-Flood discharge simulations were conducted from 1985 to 2023.

**2.3 Model Evaluation**

To quantify the performance of the different runoff schemes, the mean annual runoff bias was analysed, calculated as:

$$Bias\ (\%) = 100 \times \frac{EXP - ERA5\text{-}Land}{ERA5\text{-}Land} \tag{18}$$

where EXP represents the runoff values for each Noah-MP runoff scheme. To avoid incorrect values due to scarce or null
runoff or very high bias resulting from low ERA5-Land runoff values, total-annual ERA5-Land runoff values below 5 mm/year were masked. This mask was then applied to the corresponding values in all experimental datasets, ensuring consistency with the ERA5-Land.

For the evaluation of the impact of the different runoff schemes on discharge simulated by CaMa-Flood, the model outputs were compared with observed discharge data obtained from the Global Runoff Data Centre (GRDC) and the simulated runoff.
The evaluation began with visualizing the temporal dynamics of runoff and discharge through mean seasonal cycle plots. Next, we analysed the runoff bias as calculated using Eq. (18) and compared it with the corresponding discharge bias. Finally, a comprehensive assessment of the model performance for each runoff scheme in terms of daily discharge was conducted using several statistical metrics, including the correlation coefficient (R), standard deviation (SD), mean absolute error (MAE), root mean squared error (RMSE), and the Kling-Gupta Efficiency (KGE). The KGE metric (Gupta et al., 2009) is an aggregated
measure of the agreement in timing, magnitude, and variability between simulations and observations. It ranges from -∞ to 1, this latter representing the perfect score. The Taylor diagram (Taylor, 2001), which allows simultaneous evaluation of the temporal correlation and standard deviations, was also used to visually summarise the performance over different climatic regions.

The comparative analysis was conducted over 43 global river basins, where discharge observations are available, spanning

four climatic zones classified according to the Köppen climate classification (Kottek et al., 2006), which are: cold, warm temperate, equatorial, and arid regions (Fig. 1). The drainage area of the selected basins, based on the discharge gauge stations, ranges from 16,920 km² for the smallest to 4,671,462 km² for the largest. This ensures that the evaluation of both Noah-MP and CaMa-Flood simulations accounts for a wide range of hydrological and climatic conditions, enhancing the robustness and generalisability of the findings. Other details regarding the dominant soil textures, land cover types, and slope characteristics

across the basins are provided in the Supplement (Table S1 and Text S1).

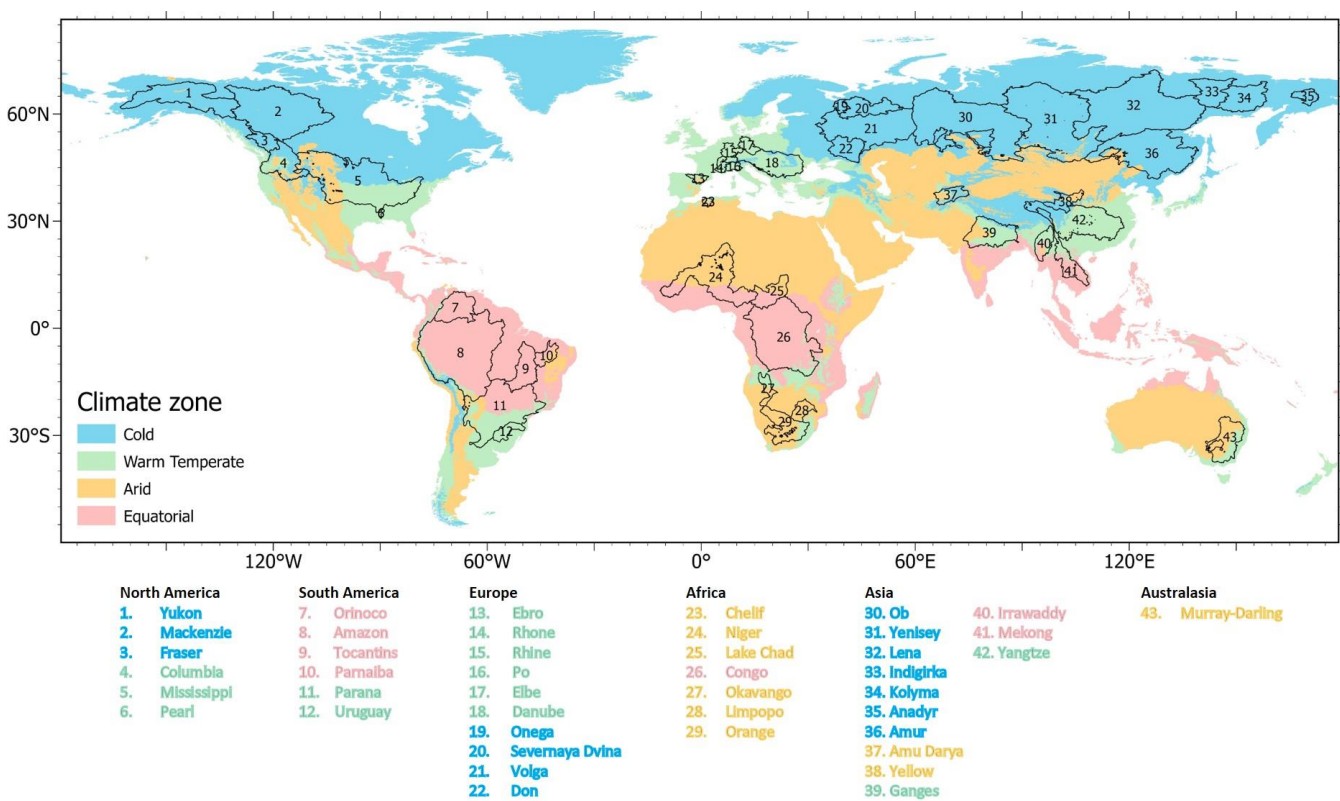

**Figure 1: Geographical locations and names of global river basins used in this study, over four climate zones based on the Köppen-Geiger climate classification world map (Kottek et al., 2006). The names of the basins are color-coded based on their respective climate zones (Cold, Warm Temperate, Arid, Equatorial). Basins spanning multiple climate zones are assigned to the zone covering**

**the largest area.**

## 3 Results and discussion

### 3.1 Runoff Bias

Figure 2 illustrates the spatial pattern of the annual mean (1985-2003) runoff biases of various Noah-MP runoff schemes compared to ERA5-Land runoff data, expressed as percentages. The biases were calculated as in Eq. (18), where positive

values indicate overestimation and negative values indicate underestimation of runoff by the Noah-MP experiments relative to ERA5-Land.

In addition, the ERA5-Land runoff map serves as reference, showing the total annual runoff in mm per year. It clearly distinguishes dry areas with runoff of less than around 140 mm per year, mostly corresponding to arid regions, from humid and very humid areas. In equatorial regions, annual runoff can exceed 1000 mm, as seen in the Amazon Basin and parts of

south-eastern Asia.

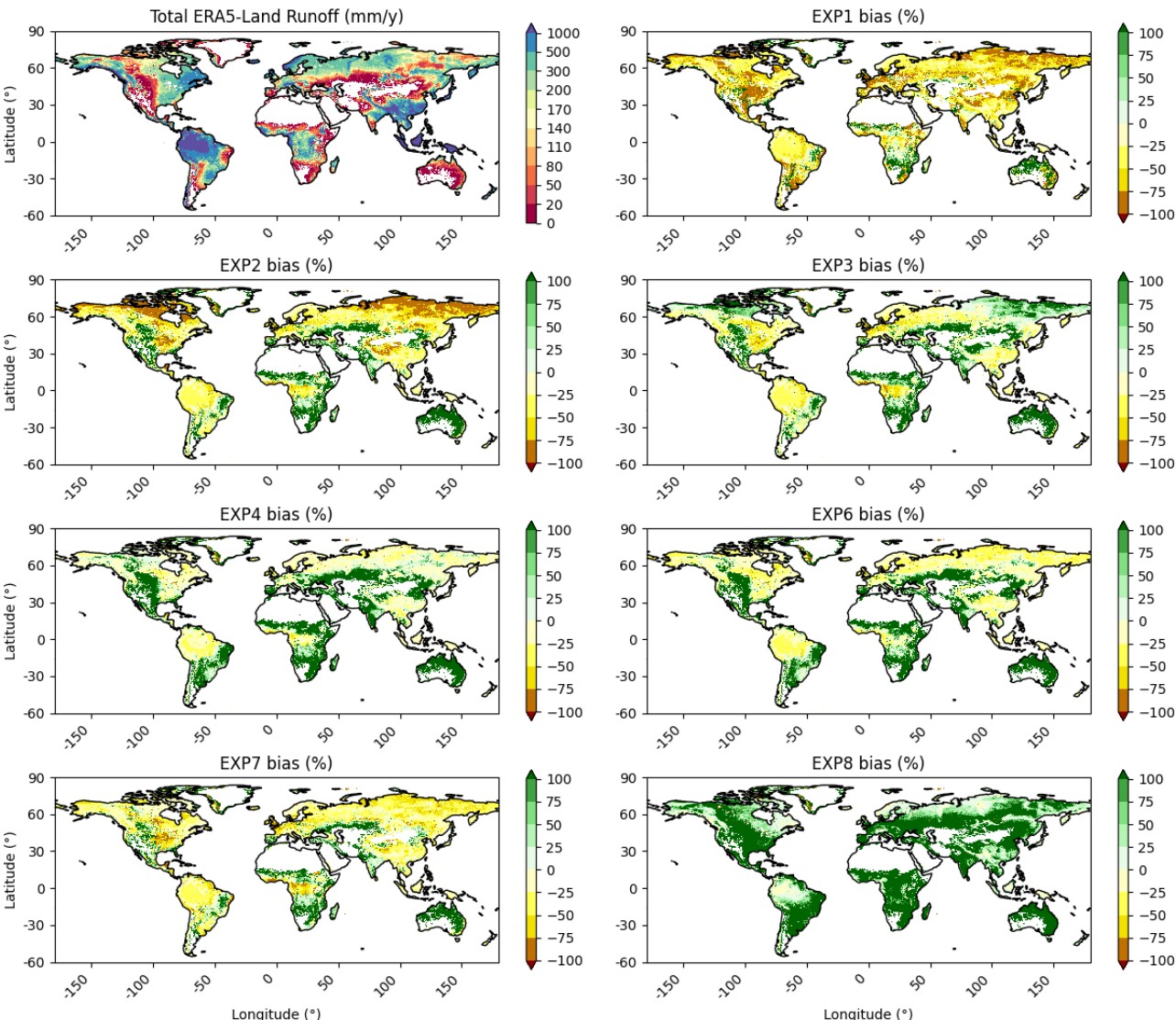

**Figure 2: Multi-year mean (1985-2023) annual runoff bias (%) of Noah-MP runoff schemes driven by ERA5-Land climate forcing compared to ERA5-Land runoff data.**

Across the different experiments, the biases in runoff exhibit substantial spatial variability. In particular, EXP1 generally underestimates runoff compared to ERA5-Land in almost every region, except in central and South Africa, and some limited areas in Eastern South America and Western Australasia. This underestimation is particularly significant in the Northern Hemisphere, especially in cold and warm temperate regions, where the bias often exceeds 50%, reaching up to 100% (~300 mm/year). This indicates that EXP1 struggles considerably with capturing runoff dynamics in these areas.

EXP2 exhibits a more balanced bias distribution: while negative biases are still present in the Northern Hemisphere, particularly in Canada and northern Asia, the magnitude is reduced compared to EXP1. There are also slightly more areas with positive bias reaching up to 400% (~50 mm/year), in Africa, Australia, and North and Eastern South America, while negative biases, exceeding 20% (~300 mm/year), are observed in the Amazon basin and Southeast Asia.

EXP3 demonstrates reduced biases overall compared to EXP1 and EXP2. In the Southern Hemisphere, its performance is close to EXP2. In the Northern Hemisphere, positive biases, lower than 100% (~150 mm/year), are observed in polar and cold regions. Negative biases are still present but are less intense. This suggests an improved handling of runoff dynamics in these regions.

EXP4 exhibits a similar pattern of bias to EXP3, with a generally lower magnitude of bias compared to the earlier experiments. However, there are still regions with significant positive bias reaching up to around 1500% (~500 mm/year), particularly in the equatorial and subequatorial zones.

EXP6 shows a mix of positive and negative biases, but with a generally lower magnitude compared to the earlier experiments. Negative biases, mostly below 50% (equivalent to less than 150 mm/year), are particularly evident in the Northern Hemisphere, especially in cold and warm temperate regions, and can reach from 200 to 400 mm/year in equatorial regions like the Amazon River basin and parts of Central Asia. Positive biases are also observed in the equatorial and subequatorial zones, ranging from 1 to around 300% (~50 to 200 mm/year). However, the extent and intensity of these biases are reduced compared to EXP1 and EXP2, suggesting an improved performance in handling runoff, though not as well as EXP3 and EXP4 in some regions.

EXP7 displays a similar pattern to EXP6, with reduced biases compared to EXP1 and EXP2. Negative biases dominate in the Northern Hemisphere, particularly in high-latitude and warm temperate regions. Positive biases are less widespread but are still present in some equatorial and subequatorial regions, including parts of South America and Africa, Central Asia and Australasia. The overall bias magnitudes are lower than those observed in EXP1 and EXP2, indicating a better performance, though still showing room for improvement in specific regions.

EXP8 displays the most pronounced positive biases, exceeding 100% across nearly all regions (over 300 mm in cold regions and up to 500 mm in other basins). This indicates a substantial overestimation of runoff by this scheme on a global scale. In Dynamic VIC (EXP8), the saturation excess runoff is conceptualised similarly to VIC (EXP6), which performs relatively well compared to ERA5-Land runoff. This suggests that the overestimation is primarily due to the parameterization of the infiltration-excess runoff within Dynamic VIC. With the Dynamic VIC scheme, Noah-MP uses three infiltration measurement methods: Philip (Philip, 1987), Green-Ampt (Heber Green and Ampt, 1911), and Smith-Parlange (Smith and Parlange, 1978) infiltration schemes, with the Philip scheme being used in this study. Improving runoff simulation performance with the

Dynamic VIC scheme could be achieved by selecting the most appropriate infiltration scheme and optimising its parameterization.

330    In summary, the progression from EXP1 through EXP7 shows a trend of decreasing bias magnitudes and improved performance in simulating runoff dynamics. While EXP1 and EXP2 exhibit significant underestimation in many regions, particularly in the Northern Hemisphere, the other experiments like EXP3, EXP4, EXP6, and EXP7 demonstrate progressively better performance with reduced biases. Nonetheless, challenges remain, particularly in accurately capturing runoff in the equatorial and subequatorial areas, as well as in certain high-latitude regions. This analysis underscores the need for ongoing

335    model refinement and calibration to enhance the predictive accuracy of runoff simulations across diverse climatic regions.

### 3.2 Seasonal Cycle of Runoff and River Discharge

The seasonal cycle of both runoff and discharge, as simulated by the Noah-MP experiments and the CaMa-Flood model, reveals significant variability across different climatic zones, highlighting the diverse hydrological processes within each region (Fig. 3). The inclusion of discharge data alongside each corresponding runoff plot emphasises the connection between

340    these two variables and underscores the crucial role of runoff in shaping the discharge patterns observed in various basins.

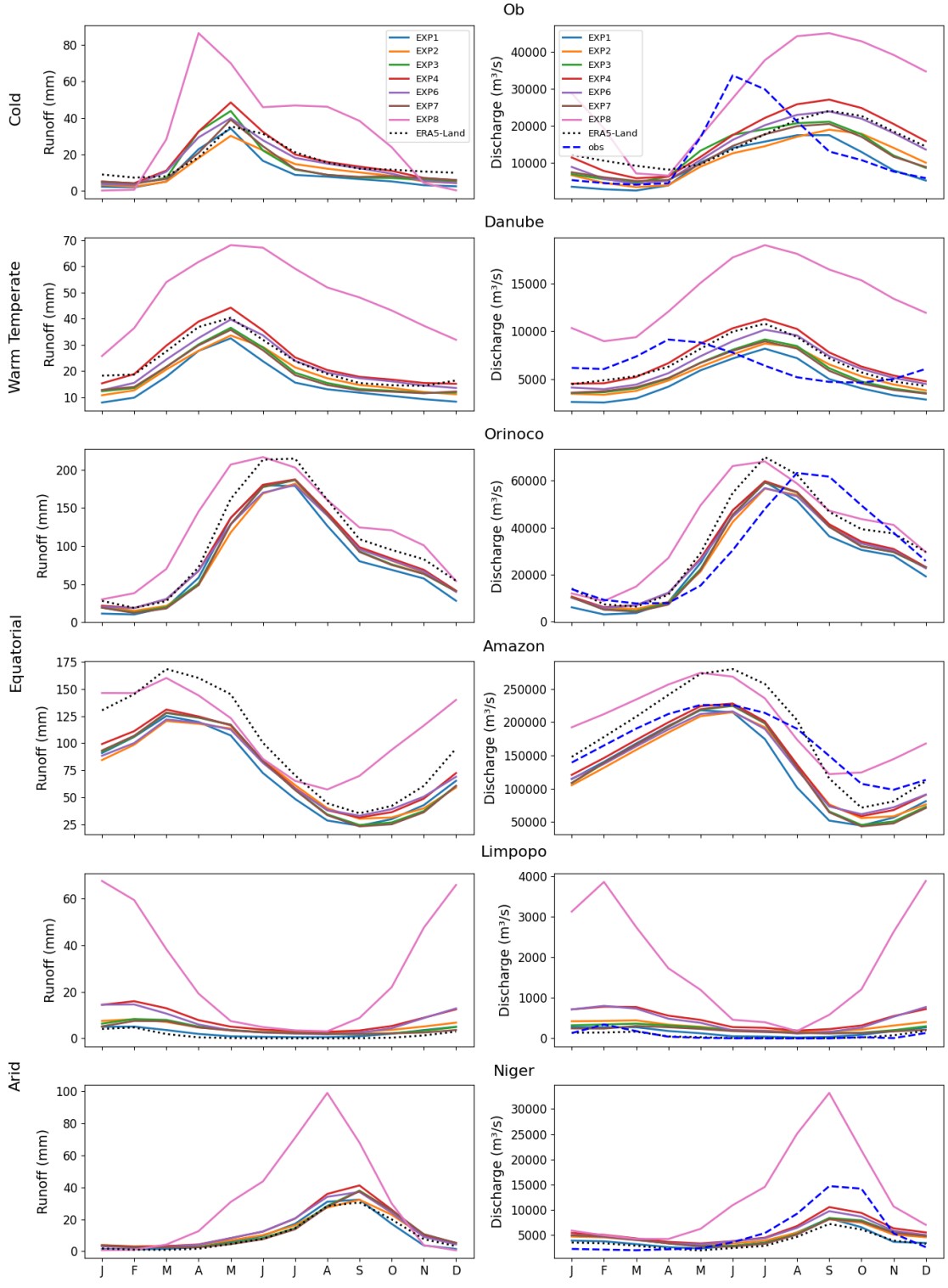

**Figure 3: Mean seasonal cycle of runoff (mm) and river discharge (m3/s) simulated by the different Noah-MP runoff schemes and CaMa-Flood, for 6 selected river basins representing four climate regions (cold, warm temperate, equatorial and arid). Discharge data includes simulated and observed values (obs) for the period 1985–2023. Observation years contributing to the monthly mean vary depending on their availability, with a minimum of 5 years per catchment.**

In equatorial regions, such as the Amazon and Orinoco basins, the seasonal cycle of runoff exhibits pronounced peaks during wet seasons, directly translating into high discharge values. These peaks are expected due to intense rainfall and the vast drainage areas characteristic of these regions. The ability of the models to replicate these extreme events is essential for understanding and managing water resources in areas prone to significant seasonal variations.

In contrast, arid regions, including the Niger and Limpopo basins, display more subdued seasonal cycles of both runoff and discharge. This pattern reflects the typically low and irregular rainfall in these regions, leading to lower and less variable runoff and, consequently, discharge. The consistency between runoff and discharge patterns in these regions illustrates the sensitivity of the models to regional climatic conditions.

Across many basins, the seasonal cycles of runoff and discharge generally agree. However, a noticeable lag often exists between the peak runoff and peak discharge, especially in large river basins like the Amazon. This lag, which can extend up to three months (Liang et al., 2020; Sorribas et al., 2020), is due to the natural routing process within the river network. This process involves the time it takes for water to travel through the system and the storage effects within river channels, depending on basin characteristics such as size, shape, drainage density, river length, and slope. In some cases, this lag could also reflect limitations in the CaMa-Flood routing model, particularly for large-scale river basins where routing dynamics are complex. A detailed sensitivity analysis of the routing parameterisation (such as river velocity, roughness coefficients, or floodplain dynamics) could offer valuable insights into how model-specific limitations impact the timing of peak discharge. This could be an important direction for future research, with the potential to enhance model performance in accurately simulating discharge timing.

In certain cases, such as the Ob and Danube river basins, the delay in capturing the peak discharge by comparison to observed peaks seems to stem from limitations in specific parametrisations and subsurface runoff scheme within the CaMa-Flood. These limitations, hinder the model's ability to accurately predict the timing of maximum discharge, which is critical for applications such as flood forecasting and water resource management. Supporting this, another river routing model, WRF-Hydro (Gochis et al., 2020), was tested using the same inputs (land surface model, schemes, forcing, resolution, and topography) for the Danube River basin. Unlike CaMa-Flood, WRF-Hydro was able to capture the peak discharge timing accurately (not shown), confirming that the delay observed in CaMa-Flood simulations is due to the model's inherent limitations and thus not linked to the Noah-MP representation of runoff.

Both ERA5-Land and Noah-MP runoff-driven discharge successfully replicate the general patterns of the mean seasonal discharge cycle across most basins and climatic regions. This indicates a robust performance of the models in capturing the overall seasonal dynamics of river discharge. This capability is advantageous for conducting trend and frequency analysis under climate change projections and scenarios, aiding in the development of water management strategies and climate actions.

Additionally, these models can be used in conjunction with a set of climate models to build an ensemble model, effectively addressing the uncertainty in future projections.

Conversely, a significant positive bias is observed in EXP8, which tends to overestimate the seasonal cycle of discharge in the majority of the basins globally. This overestimation is evident when compared to ERA5-Land runoff-driven discharge as well
as to the other seven experimental setups. The consistent overestimation by EXP8 confirms the systematic issue within this particular runoff scheme. On the other side, EXP1 and EXP2 slightly underestimate discharge when compared to ERA5-Land in cold, warm temperate, and equatorial regions. However, these schemes slightly overestimate discharge in arid regions. This differential performance indicates region-specific biases in these runoff schemes that might be attributed to how they handle surface and subsurface runoff processes. The noticeable bias highlighted with these experiments reflects clearly the translation
of the aforementioned bias in runoff to a bias in discharge. This shows the significant influence that the selection of a particular runoff scheme can have on flow discharge.

Finally, we would highlight that all experimental setups and ERA5-Land sometimes tend to overestimate the discharge when compared to GRDC observations (Fig. S1 in the Supplement). This overestimation can be attributed to various factors, including basin regulation and human activities, observation accuracy, and forcing accuracy and resolution. These elements
introduce complexities that affect the ability of models to match observed discharge precisely.

### 3.3  Runoff Bias Propagation to Discharge Bias

After demonstrating the significant impact that the selection of a particular runoff scheme can have on flow discharge patterns, we now explore the relationship between runoff bias and discharge bias. According to the water balance equation, within a defined area over a specific period, the total inflows (such as precipitation) must equal the total outflows (including runoff and
evapotranspiration), plus any change in storage (such as changes in soil moisture, groundwater, or surface water reservoirs). When considering periods longer than one year, the changes in water storage are generally assumed to be negligible ($\Delta S = 0$) (Oda et al., 2024). Under this assumption, the total runoff (including both surface and subsurface components) is nearly equal to the total discharge observed at the basin outlet. As a result, one would expect a strong correlation between runoff bias and discharge bias.
Figure 4 corroborates this expectation, illustrating the correlation between runoff and discharge biases when compared to ERA5-Land runoff and simulated discharge across various climate zones and globally.

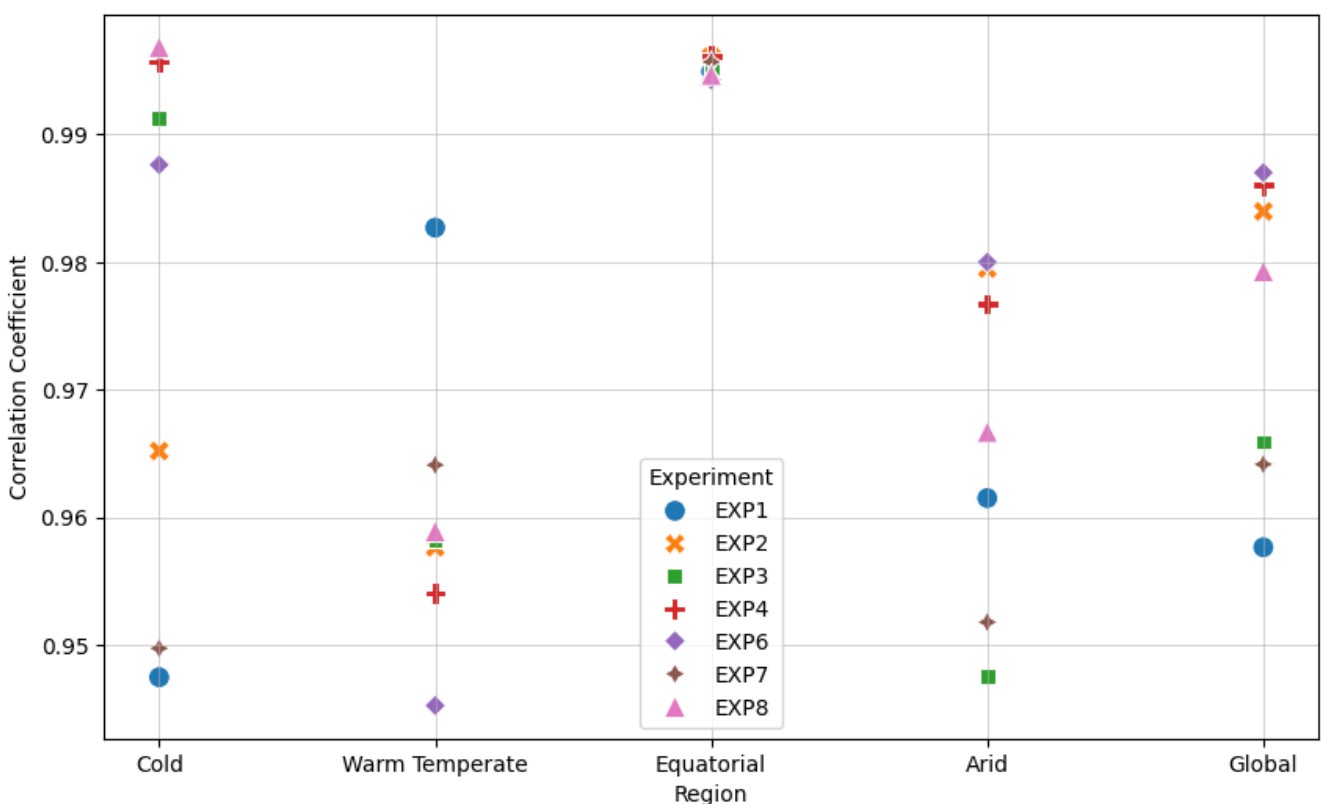

**Figure 4: Scatter plots of Pearson correlation coefficients between runoff and discharge biases across climate regions and globally.**

The correlation coefficients, ranging from 0.945 to 0.997, indicate an almost perfect agreement between the biases,
highlighting the direct propagation of uncertainty from runoff to discharge estimation. This analytical finding aligns with the
results of David et al. (2019), further emphasizing the critical influence that the choice of runoff scheme has on discharge bias.

### 3.4 Discharge Performance Metrics

Table 2, Table 3 and Fig. 5 present the performance statistical metrics, i.e. the Kling-Gupta Efficiency (KGE), Root Mean
Square Error (RMSE), Mean Absolute Error (MAE), temporal correlation coefficient (R), and standard deviation (SD), of the
different Noah-MP experiments at the global scale and across the four climate regions against the observational dataset GRDC
(a detailed values regarding some of these metrics are provided for each river basin in the Tables S2, S3 and S4 in the
Supplement). These metrics provide a comprehensive evaluation of each model's performance in capturing daily discharge,
highlighting the strengths and limitations of each experimental setup globally and in diverse climatic conditions.

 **Table 2: Global Performance metrics (KGE, R, SD (m³/s), RMSE (m³/s), MAE (m³/s)) of Noah-MP runoff models in terms of daily discharge.**

| EXPs | Global | | | | |
|---|---|---|---|---|---|
| | KGE | R | SD | RMSE | MAE |
| EXP1 | 0.70 | 0.95 | 27773.7 | 11628.8 | 4246.2 |
| EXP2 | 0.75 | 0.97 | 27859.3 | 9791.5 | 3830.9 |
| EXP3 | 0.82 | 0.96 | 28905.1 | 9995.4 | 3741.6 |
| EXP4 | 0.89 | 0.96 | 30020.2 | 9200.4 | 3728.6 |
| EXP6 | 0.84 | 0.97 | 28529.4 | 9476.5 | 3675.4 |
| EXP7 | 0.78 | 0.96 | 28702.1 | 10558.2 | 3890.8 |
| EXP8 | 0.31 | 0.95 | 41170.9 | 16359.7 | 8858.9 |
| ERA5-Land | 0.87 | 0.97 | 36919.6 | 9024.9 | 3677.4 |
| Observations | | | 33361.1 | | |

Considering the Noah-MP performances in discharge simulation, at the global scale (Table 2), EXP4 shows the best performance with a KGE of 0.89, a high correlation (0.96), and reasonable error metrics. ERA5-Land also performs well with

a high KGE (0.87) and moderate error metrics. EXP3 and EXP6 show strong performance with KGEs of 0.82 and 0.84, respectively, and moderate error metrics. EXP2 and EXP1 demonstrate balanced performance with KGEs of 0.75 and 0.70, respectively. EXP7 performs moderately with a KGE of 0.78. EXP8, however, performs poorly with the lowest global KGE (0.31) and the highest SD, RMSE, and MAE, indicating significant overestimation.

**Table 3: Performance metrics (KGE, RMSE (m³/s), MAE (m³/s)) of Noah-MP runoff models in terms of daily discharge across climate regions.**

| EXPs | Cold Regions | | | Warm Temperate Regions | | | Equatorial Regions | | | Arid Regions | | |
|---|---|---|---|---|---|---|---|---|---|---|---|---|
| | KGE | RMSE | MAE | KGE | RMSE | MAE | KGE | RMSE | MAE | KGE | RMSE | MAE |
| EXP1 | 0.39 | 7578.3 | 3436.6 | 0.53 | 5716.5 | 2856.0 | 0.75 | 26212.9 | 13349.6 | 0.59 | 1241.9 | 549.2 |
| EXP2 | 0.45 | 7274.0 | 3430.5 | 0.76 | 4693.2 | 2315.4 | 0.77 | 21529.6 | 11608.6 | 0.44 | 1375.5 | 810.6 |
| EXP3 | 0.79 | 5900.8 | 2998.5 | 0.82 | 4647.9 | 2315.0 | 0.80 | 22992.3 | 11982.4 | 0.47 | 1344.0 | 759.8 |
| EXP4 | 0.62 | 7290.8 | 3290.9 | 0.64 | 4922.2 | 2544.7 | 0.86 | 19576.9 | 10288.8 | -0.11 | 1717.9 | 1171.4 |

| | | | | | | | | | | | | |
|---|---|---|---|---|---|---|---|---|---|---|---|---|
| EXP6 | 0.57 | 7247.5 | 3049.7 | 0.79 | 4532.0 | 2367.9 | 0.81 | 20675.1 | 11027.0 | 0.13 | 1564.0 | 1018.7 |
| EXP7 | 0.45 | 7824.1 | 3342.8 | 0.78 | 4734.0 | 2369.2 | 0.80 | 23399.1 | 12171.5 | 0.55 | 1330.1 | 710.0 |
| EXP8 | 0.12 | 11396.1 | 7089.8 | -0.92 | 15244.6 | 8147.3 | 0.62 | 29853.7 | 19879.7 | -4.14 | 6512.5 | 3836.1 |
| ERA5-Land | 0.55 | 8165.0 | 3964.2 | 0.87 | 3881.9 | 2003.5 | 0.84 | 18892.0 | 10555.2 | 0.52 | 1317.0 | 514.4 |

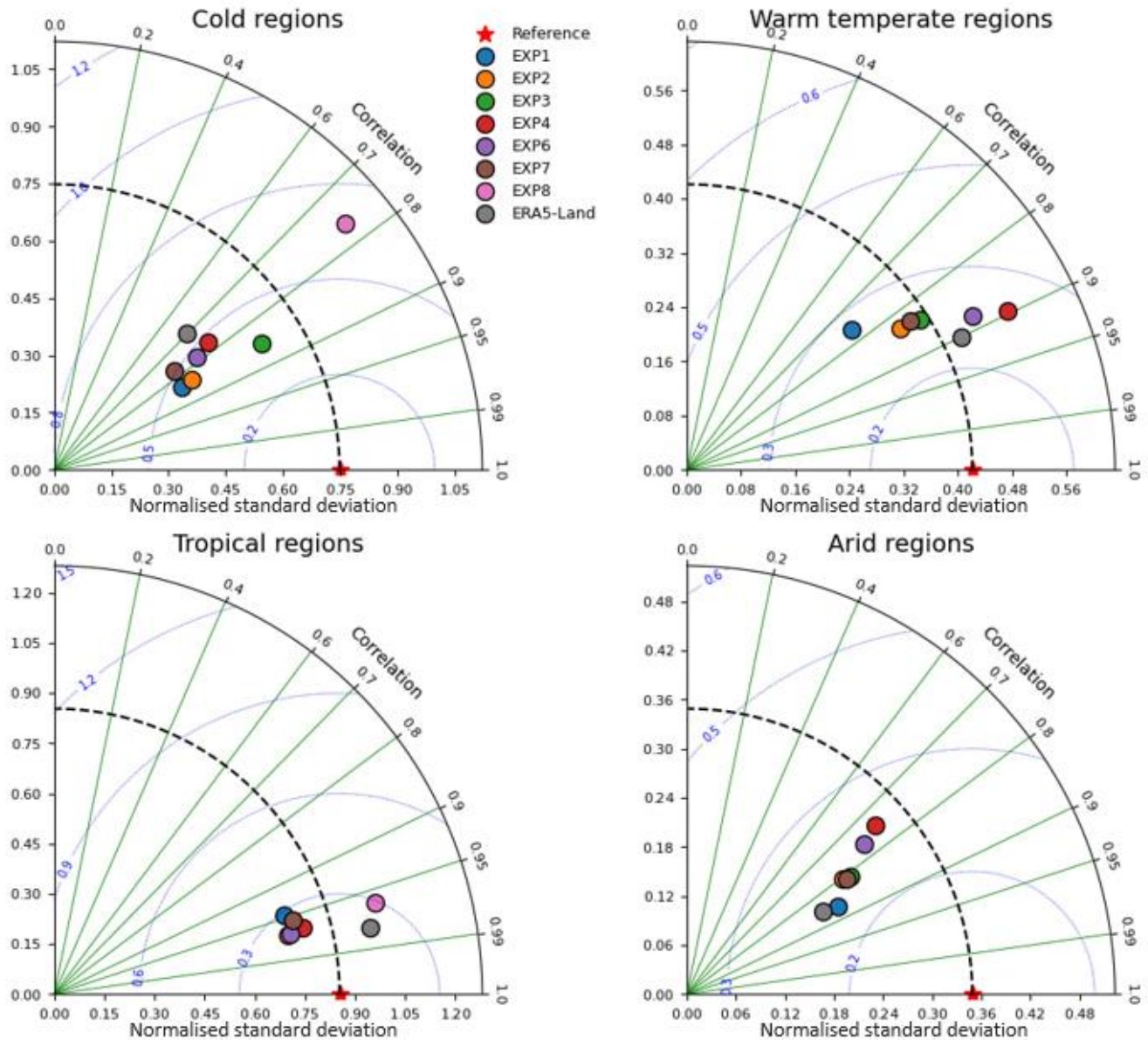

**Figure 5: Taylor diagram showing the performances of Noah-MP runoff models in terms of daily discharge within four climate regions.**

In the cold regions, EXP3 exhibits the highest KGE (0.79) and a high correlation coefficient (0.86), closely matching the observed standard deviation (SD). This model also has the lowest RMSE (5900.8 m3/s) and MAE (2998.5 m3/s), indicating strong performance. EXP4 and EXP6, as well as ERA5-Land, also perform well with KGEs of 0.62, 0.57 and 0.55, respectively, and reasonable error metrics. EXP1 and EXP2 perform moderately, with EXP1 showing a lower SD compared

to observations, indicating an underestimation of discharge variability. EXP8 performs poorly, with the lowest KGE (0.12) and the highest SD, RMSE, and MAE, indicating significant overestimation.

In warm temperate regions, ERA5-Land achieves the best performance with the highest KGE (0.87) and correlation coefficient (0.90), closely matching the observed SD and having the lowest RMSE (3881.9 m3/s) and MAE (2003.5 m3/s). EXP3 also performs well with a KGE of 0.82 and low error metrics, followed by EXP6 and EXP7, both demonstrating strong performance.

EXP2 shows balanced performance with a KGE of 0.76 and moderate error metrics. EXP1 performs moderately with a KGE of 0.53. EXP4, despite having a high correlation (0.90), shows overestimation with a higher SD and moderate errors. EXP8 again performs poorly with the lowest KGE (-0.92) and significantly high SD, RMSE, and MAE. Due to its exceptionally high standard deviation, which is 2.4 times higher than the observed SD, EXP8 is considered an outlier in the Taylor diagram and does not appear in Fig. 5. For reference, its correlation coefficient is 0.88.

In equatorial regions, EXP4 and ERA5-Land show the highest performance with KGEs of 0.86 and 0.84, and a high correlation (0.97 and 0.98), respectively. These models closely align with the observed SD and exhibit the lowest RMSE (19576.9 m3/s and 18892.0 m3/s) and MAE (10288.8 m3/s and 10555.2 m3/s). EXP2, EXP3, EXP6 and EXP7 also perform well with KGEs of 0.77, 0.80, 0.81 and 0.80, respectively, and present moderate error metrics. EXP1 shows moderate performance, while EXP8 performs poorly with the highest SD, RMSE, and MAE.

In arid regions, ERA5-Land again shows good performance with a KGE of 0.52 and a high correlation (0.85), closely matching the observed SD. EXP1 and EXP7 also perform reasonably well, with KGEs of 0.59 and 0.55, respectively. EXP2 and EXP3 show balanced performance with moderate KGEs. EXP6 shows a lower performance with a KGE of 0.13 and higher error metrics. EXP4 and EXP8 perform poorly, with EXP8 having the lowest KGE (-4.14) and the highest error metrics, indicating significant overestimation. The high standard deviation of EXP8, which is almost three times higher than the observed SD,

makes it an outlier in the Taylor diagram, hence it is not plotted. However, its correlation coefficient is 0.58.

Overall, ERA5-Land and Schaake approach (EXP3) consistently exhibit strong performance across different climate regions, closely aligning with observed data and demonstrating low error metrics. Other models like VIC and BATS (EXP 6 and 4) also perform well in specific regions, while Dynamic VIC (EXP8) consistently underperforms, showing substantial overestimation and high error metrics due to the strong overestimation in runoff.

The results presented are not surprising and are, in fact, reasonable given the characteristics of the different runoff schemes and the regions they are applied to. Schaake, an infiltration-excess runoff scheme (EXP3), performs the best in the northern mid-and-high latitude basins as stated in (Decharme, 2007), here corresponding to cold regions, i.e. areas dominated by snow and glaciers, which make the frozen soils less permeable. Additionally, the intensity of precipitation in these regions often exceeds the infiltration rate, thereby generating runoff through the infiltration-excess mechanism.

EXP3 also performs exceptionally well in warm temperate regions, which dominate the CONUS (Continental United States) domain. This superior performance likely justifies its status as the default option in WRF-Hydro/US National Water Model.

Schaake, along with BATS, VIC, and XAJ (i.e. respectively EXP3, 4, 6 and 7), shows strong performance in warm temperate and equatorial regions. This aligns with existing literature indicating that these schemes were developed and tested primarily in humid and sub-humid regions (Hao et al., 2015; Hou et al., 2023).

In arid regions, TOPMODEL with groundwater (EXP1) stands out, likely due to its incorporation of subsurface and groundwater processes, which enhance groundwater recharge and baseflow contributions—factors that are crucial in arid environments. The same reason could be behind its underestimation of runoff in humid and sub-humid regions. A study by (Gan et al., 2019) demonstrated that TOPMODEL with groundwater scheme produces the wettest soil and the greatest evapotranspiration across ten hydrologic regions of China, in contrast to the BATS scheme, which yields the driest soil and

the smallest evapotranspiration.

ERA5-Land demonstrates very good performance at both global and regional scales, serving as an excellent reference dataset for runoff benchmarking.

The hydrological basins were also grouped based on dominant soil texture and land cover types to analyse the performance of the Noah-MP model. Additionally, the experiments' performances were correlated with the mean slope, as well as the basin

size (Fig. S2 and Text S2 in the Supplement). No significant correlations were found overall, indicating that the Noah-MP model operates effectively regardless of land cover type, soil type, basin size, or topography. This lack of correlation suggests a robust versatility in the model's application, highlighting its capacity to provide reliable simulations across diverse environmental conditions and varied landscape features. The findings reinforce the model's utility in different hydrological contexts, supporting its use in global and regional hydrological studies without the need for extensive customisation based on

specific basin characteristics.

Although in general, except for Dynamic VIC, the experiments demonstrated good performance in capturing the overall patterns of river discharge, there remains a considerable bias when applied to more detailed studies, particularly those focusing on high-flow extremes. This bias affects the accuracy in capturing the magnitude, timing, and extent of these events, indicating that further improvements are necessary.

The differences in biases between the runoff schemes are driven by how each scheme handles critical hydrological processes, particularly the partitioning of precipitation into surface and subsurface runoff, as well as the treatment of soil moisture dynamics. Although the formulas for some runoff schemes (Table 1) appear similar, the variations in specific parameters and their physical representations cause distinct biases to emerge.

For example, in the TOPMODEL-based schemes, while the surface runoff follows the same formula across both experiments,

subsurface runoff differs in the way it is computed. In EXP1, subsurface runoff is influenced by the soil hydraulic conductivity of the first unsaturated layer above the water table, while in EXP2, the subsurface runoff is controlled by a fixed base flow coefficient and a fixed runoff decay factor. These differences lead to distinct sensitivities in how each scheme responds to variations in soil properties and terrain. The water table depth also plays a pivotal role in the calculation of subsurface flow in these schemes, introducing differences in regions with shallow or deep groundwater. These variations affect the magnitude

and timing of runoff, which ultimately manifests as bias in the discharge simulation.

The surface runoff in the TOPMODEL-based schemes and BATS also shares the same formula; however, the differences lie in how the saturated area fraction (fsat) is calculated. In the TOPMODEL-based schemes, fsat depends on the water table depth and a runoff decay factor, which differs between the two schemes, while in BATS, it is computed as the fourth power of the degree of saturation in the top two meters of soil. This distinction between the two approaches introduces variability in how surface runoff is generated across regions with differing soil moisture profiles and saturation conditions. The BATS scheme also handles subsurface runoff differently, using a free drainage approach where it is calculated as the product of soil hydraulic conductivity and (1 - $f_{imp,max}$). This method leads to a distinct response to soil permeability and introduces varying biases depending on the frozen or compacted soil conditions.

For the other free-drainage schemes, including Schaake, VIC, XAJ, and Dynamic VIC, the calculation of subsurface runoff follows a similar approach. However, the key differences between these schemes lie in how surface runoff is generated. In the Schaake scheme, surface runoff is governed by an infiltration-excess mechanism, which depends on the total maximum holdable soil water content and the rate at which the soil can infiltrate water. This mechanism tends to produce lower biases in regions where infiltration-excess processes dominate, such as cold regions (Decharme, 2007).

The VIC scheme, on the other hand, calculates surface runoff based on the infiltration and maximum infiltration capacity of the soil, which introduces a different partitioning of rainfall into surface and subsurface components. The scheme's reliance on current soil moisture conditions, particularly in the tension water storage in the top layers of the soil, leads to varying biases depending on whether the region is experiencing wet or dry conditions. Similarly, XAJ introduces a unique approach by using a shape parameter to calculate surface runoff, which adjusts runoff generation based on the catchment's topographic and moisture characteristics. This leads to differences in performance depending on the terrain and hydrological profile of the region.

Dynamic VIC incorporates both infiltration-excess and saturation-excess runoff, further complicating the balance between surface and subsurface flow. The detailed soil moisture capacity parameters used in this scheme contribute to its dynamic nature but also make it more sensitive to inaccuracies in modelling infiltration and saturation, leading to large biases in discharge performance. The different ways each scheme handles these physical processes—whether through the treatment of soil moisture, the representation of surface and subsurface interactions, or the response to topographic and climatic variability—accounts for the differences in bias observed across the experiments. Understanding these physical distinctions is essential for improving the accuracy of runoff and discharge simulations, especially in regions with complex hydrological behaviour.

Each runoff scheme, with its unique conceptual framework, involves a set of tuneable variables and parameters, such as: soil depth, maximum surface saturated fraction ($fsat_{max}$), saturated value of soil moisture and others summarised in Table S5 in the Supplement. An area for improvement would involve calibrating these parameters, particularly at finer resolutions, to more precisely simulate runoff behaviour across diverse regions. For example, $fsat_{max}$ is often set as a global mean, but recent studies, such as Zhang et al. (2022), illustrate that using spatially variable values informed by remote sensing data (e.g., GIEMS-2) could yield more accurate regional simulations. Similarly, the fixed soil depth used in each scheme could be improved through

further spatially variable parametrisation within Noah-MP, which may help modulate runoff according to regional soil profiles and enhance the model's representation of subsurface flow dynamics. By exploring such refinements, future applications of these models could achieve better performance, especially when simulating high-flow events critical for flood risk assessments and water resource management.

We would like to underline that the obtained results and this study are biassed and constrained by the availability of high-quality discharge observations within the considered study period. This limitation, along with the need for a consistent evaluation period and the focus on less regulated basins, conditions the number of basins selected and their distribution within each climate region.

### 3.5 Implications for Global Discharge Simulation

This study provides a comprehensive analysis of how distinct Noah-MP runoff schemes impact discharge simulation in a global hydrological context, paving the way for enhanced modelling accuracy across different climate zones. By revealing the performance variability of different runoff schemes—such as Schaake, BATS, VIC, and XAJ—across cold, warm temperate, tropical, and arid regions, this research suggests that tailored scheme selection could improve discharge simulations for specific hydrological conditions. For instance, Schaake, BATS, and VIC exhibited reduced biases in warm temperate and tropical regions, while TOPMODEL-based schemes with groundwater performed notably better in arid areas, underscoring the need for strategic scheme selection based on regional climate and hydrological characteristics. This targeted approach to scheme selection can minimise bias and enhance model reliability in both regional and global discharge simulations, improving the accuracy of water resource management.

This study also addresses a crucial challenge in hydrological modelling: the significant biases in high-flow extremes by certain schemes. Given that accurate high-flow discharge predictions are essential for flood forecasting and disaster management, this finding suggests an urgent need for refining high-flow calibration. This enhancement is particularly relevant for global flood risk management, as it enables more reliable flood predictions that are vital for preparing for extreme weather events. By identifying critical parameters and proposing spatially variable adjustments, such as using data from sources like remote sensing products, this study sets a practical foundation for developing global calibration strategies that could yield more accurate discharge predictions (Beck et al., 2017). These strategies could be applied universally across a range of climates, creating a more adaptable global model without extensive customization.

This research further advances current hydrological modelling by demonstrating the value of multi-model comparisons, which allow for a holistic approach to discharge simulation. Rather than depending on a single runoff scheme and potentially inheriting its limitations (Diks and Vrugt, 2010; Shoaib et al., 2018), a multi-scheme approach enables researchers to capture river discharge dynamics more comprehensively (Georgakakos et al., 2004, Huo et al., 2019). This approach aligns with a broader hydrological perspective that considers the interactions between multiple runoff dynamics, offering a pathway for more nuanced simulations that acknowledge the strengths and limitations of each scheme.

For coupled ocean-atmosphere regional models lacking complete river and discharge representations, integrating findings from this study could significantly improve their hydrological modules, particularly in complex regions like the Mediterranean where the freshwater flux from rivers remarkably affects the salinity near the coast close to river mouths (e.g. Reale et al., 2020). These refinements are expected to enhance the overall representation of the global water cycle within climate models, providing more realistic freshwater flux predictions and supporting more accurate climate projections.

Additionally, this study's analysis of seasonal and regional discharge cycles reveals new insights into the variability of discharge patterns across climates. This detailed understanding could facilitate the development of models better suited to capture seasonal dynamics in tropical and temperate regions, where runoff schemes like Schaake and ERA5-Land-driven simulations performed particularly well. By capturing the discharge seasonality more accurately, our findings have direct applications for both short- and long-term forecasting and water resources planning (Pires and Martins, 2024), especially in regions facing pronounced seasonal changes in water availability.

Furthermore, the findings related to groundwater interactions underscore the importance of accurate groundwater dynamics in discharge simulation, especially in arid regions. The effectiveness of TOPMODEL-based schemes with groundwater dynamics in these areas suggests that future modelling efforts should prioritise improving groundwater parameterizations, particularly where groundwater plays a critical role in maintaining streamflow. This refinement could improve discharge simulations (Decharme and Colin, 2024), especially in water-scarce areas, supporting more efficient resource allocation and resilience against drought.

Finally, the implications of these findings extend to climate adaptation strategies, where reliable hydrological models are critical for anticipating shifts in water availability under changing climates. By advancing the accuracy of discharge simulations, particularly in high-flow and seasonal scenarios, this research provides a basis for better-informed adaptation planning, enabling decision-makers to prepare for anticipated changes in river flow and water availability. This study not only enhances current global hydrological modelling but also lays a foundation for more resilient water resource management, which is increasingly critical as climate variability challenges water availability worldwide.

## 4 Conclusions

This study evaluated the performance of seven different Noah-MP runoff schemes in discharge simulations, as simulated using the CaMa-Flood River routing model. Using ERA5-Land runoff data as a benchmark for runoff evaluation and streamflow observations for discharge evaluation across various climatic regions, key findings from the analysis reveal significant differences in how each scheme handles runoff dynamics. These findings have important implications for global hydrological modelling and water resource management.

The progression from TOPMODEL-based schemes through Schaake, BATS and other saturation-excess schemes showed a trend of decreasing bias magnitudes and improved performance in simulating global runoff dynamics. TOPMODEL with groundwater and TOPMODEL with an equilibrium water table significantly underestimated runoff in many regions,

particularly in the Northern Hemisphere, while runoff schemes like Schaake, BATS, VIC, and XAJ demonstrated progressively better performance with relatively lower biases. Dynamic VIC consistently overestimated runoff across nearly all regions.

Seasonal cycle analysis using CaMa-Flood driven by different Noah-MP runoff schemes highlighted considerable regional and seasonal variability in discharge patterns. ERA5-Land runoff-driven discharge and several Noah-MP experiments successfully replicated the general patterns of mean seasonal discharge cycles across diverse river basins. However, Dynamic VIC showed a significant positive bias, indicating a tendency to overestimate discharge globally, due to the strong runoff overestimation.

Globally, our findings reveal that EXP4 offers the best performance in discharge simulation, achieving the highest KGE, strong temporal correlation, and balanced error metrics. This indicates its robust applicability for capturing the daily discharge dynamics on a global scale. ERA5-Land and other models such as Schaake and VIC also demonstrate solid performance, particularly in regions with distinct hydrological characteristics.

Regionally, ERA5-Land and Schaake scheme consistently exhibited strong performance across different climate regions, closely aligning with observed data and demonstrating low error metrics. In contrast, TOPMODEL and Dynamic VIC showed higher error metrics, with more significant biases for Dynamic VIC, indicating the need for further refinement, although TOPMODEL with groundwater stands out as the most effective in arid regions.

The Noah-MP model demonstrated robust versatility, performing effectively regardless of land cover type, soil type, basin size, or topography. This suggests that the model can provide reliable simulations across diverse environmental conditions without extensive customisation.

While the experiments generally captured the overall patterns of river discharge, significant biases remained, particularly in high-flow extremes. This underscores the need for ongoing calibration of tuneable variables and parameters, especially at finer resolutions, to enhance the accuracy and reliability of hydrological simulations.

In conclusion, this study transcends the limitations of individual schemes and specific regions, providing a holistic assessment of runoff dynamics on a global scale. The analysis underscores the significant impact that the selection of a particular runoff scheme can have on discharge patterns and bias, emphasizing the necessity for careful scheme selection based on specific hydrological contexts. Enhanced calibration and refinement efforts are essential for achieving more accurate hydrological predictions, which are vital for effective water resources management and climate adaptation strategies across diverse global environments.

**Code and data availability**

GRDC discharge observations can be obtained from https://portal.grdc.bafg.de/applications/public.html?publicuser=PublicUser#dataDownload.

ERA5-Land data, provided by the European Centre for Medium-Range Weather Forecast (ECMWF), can be freely downloaded from the Copernicus Data Store (https://cds.climate.copernicus.eu/cdsapp#!/dataset/reanalysis-era5-land?tab=overview).

Noah-MP and CaMa-flood models can be downloaded from https://github.com/NCAR/noahmp and https://github.com/global-hydrodynamics/CaMa-Flood_v4, respectively.

Enquiries about output data availability should be directed to the authors.

## Author contribution

MH, GF, AA and CH: conceptualisation and methodology; MH: formal analysis, funding acquisition, investigation, visualisation and writing-original draft preparation; MH and AA: data collection, curation and simulations; GF and AA: supervision; GF, AA, CH and TL: validation and writing-review & editing. All authors have read and agreed to the published version of the paper.

## Competing interests

The authors declare that they have no conflict of interest.

## Acknowledgements

We would like to thank Dr. Stefan Hagemann and Dr. Stephen Birkinshaw for providing us with some discharge observation data and Dr. Day Yamazaki for his valuable help in setting up the CaMa-flood model.

## Financial support

This paper and related research have been conducted during and with the support of the Italian PhD course in Sustainable Development and Climate change (link: www.phd-sdc.it) at the University School for Advanced Studies IUSS and developed within the framework of the project "Dipartimento di Eccellenza 2023-2027"., with the financial support from the ICSC Italian Research Center on High-Performance Computing, Big Data and Quantum Computing and received funding from the European Union Next-GenerationEU (National Recovery and Resilience Plan-NRRP, Mission 4, Component 2, Investment 1.4-D.D: 3138 16/12/2021, CN00000013).

This research has also been supported by the CIHEAM Prize for the Best MSc Thesis 2022.

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
