# Peer review of "Impact of Runoff Schemes on Global Flow Discharge: A Comprehensive Analysis Using the Noah-MP and CaMa-Flood Models"

_Hydrology and Earth System Sciences, 2024_

## Author Comment (AC1)

The manuscript evaluates seven runoff schemes in the Noah-MP land surface model for estimating global river discharge, comparing them to ERA5-Land data and streamflow observations. Results show varying accuracy, with TOPMODEL-based schemes underestimating runoff in some regions, while others like Schaake and BATS performed better. Dynamic VIC overestimated runoff globally. The study indicated that despite good performance, biases in high-flow extremes highlight the need for further model improvements. The study emphasizes improving hydrological models for accurate water resource management and climate adaptation.

The study addresses a quite interesting topic. The manuscript is well organized and neatly written with the appropriate scientific content. However, I have some suggestions and questions as follow.

Major comments:

1)    The paper does not adequately address how the insights from these evaluations could be used to advance global hydrological modeling, particularly in the context of discharge simulation. While it provides a thorough assessment of the performance of seven runoff schemes within the Noah-MP Land Surface Model, its contribution to improving hydrological modeling remains unclear.

- *Thank you for your valuable feedback. We agree with the reviewer, hence we have added a dedicated section "3.5 Implications for Global Discharge Simulation" to discuss how our findings contribute to improve hydrological modelling. See below.*

*3.5 Implications for Global Discharge Simulation:*

[revised manuscript text omitted]

2)    Line 358-362: It is noted that the lags between peak runoff and peak discharge in large river basins, such as the Amazon, are attributed to the natural routing lag within the river network. Could these lags also be due to specific limitations within the CaMa-Flood global river routing model? Have you conducted any sensitivity analysis on the models to explore this?

- *Thank you for raising this important point. We agree that the lags between peak runoff and peak discharge in large basins like the Amazon could potentially be influenced by specific limitations within the CaMa-Flood global river routing model, in addition to the natural routing lag within the river network.*

  *Since our current analysis primarily focused on the sensitivity to runoff schemes, we did not conduct a detailed sensitivity analysis specifically targeting the routing parameterisation in CaMa-Flood. We acknowledge that this could provide valuable insights into the influence of model-specific limitations on the timing of peak discharge, particularly in large-scale river basins where routing dynamics are more complex. Besides, part of the analysis suggested by the reviewer for this specific basin has already been performed in other studies (e.g. Yamazaki et al, 2012; https://doi.org/10.1029/2012WR011869).*

  *Thus, as this aspect was outside the scope of the current study, we suggest that a sensitivity analysis on key parameters (such as river velocity, roughness coefficients, or floodplain dynamics) within the CaMa-Flood model could be an important direction for future research. Such an analysis would help isolate the contributions of model limitations from natural routing processes and provide a clearer understanding of the observed lags.*

  *We appreciate the reviewer's suggestion and will consider it in future studies to further refine the modelling of discharge timing in large river systems.*

Minor comments:

1) Abbreviations are used in the abstract that may be unclear to readers who are not very familiar with the study.

   - *Thank you for your feedback. We have revised the abstract to define all abbreviations upon first use, ensuring clarity for readers.*

2) Line 25: Rephrase this sentence to better highlight the importance of this study, e.g.: "These findings are critical for improving global hydrological models, which are essential for developing more reliable water resource management strategies and adapting to the growing challenges posed by climate change, such as shifts in water availability and extreme flood events."

   - *We have revised the sentence incorporating the suggested phrasing.*

3) Line 39: Please use formal expression for the "On the flip side, ...."

   - *Thank you for your suggestion. We have revised the expression to a more formal tone as recommended ("On the other hand, …").*

4) Line 202: Is it correct that the ERA5-Land variables were regridded from 0.1° to 0.2°? If so, perhaps using a term like "regridding," "spatial aggregation," or "extrapolation" would be more appropriate.

- *Thank you for your suggestion. We have replaced 'interpolated' with 'regridded', which is more correct.*

5)   Line 241: In this study, ERA5-Land runoff is used as a benchmark for evaluating the runoff simulated by Noah-MP. While ideally, direct runoff observations would be used for this purpose, such data was not available, as you mentioned. To further strengthen your evaluation, and ensuring that the simulations align with real-world observations, it would be helpful to cite studies that have assessed ERA5-Land runoff against direct runoff measurements to demonstrate the reliability of ERA5-Land as a reference dataset.

- *Thank you for your valuable suggestion. We were unable to find studies that directly assess ERA5-Land runoff against direct runoff measurements. However, we have included the available reference that assesses ERA5-Land runoff against P-E and simulated runoff highlighting strength and weakness of the dataset.*

6)   Page 10, Figure 1: What was the reason behind selecting these specific river basins? Were they chosen based on their size as the largest river basins?

- *The specific river basins were selected based on their status as large river basins globally, while also being constrained by the availability of consistent data.*

7)   Line 393-394: I suggest using a more complete version of your statement something like this "According to the water balance equation, within a defined area over a specific period, the total inflows (such as precipitation) must equal the total outflows (including runoff and evapotranspiration), plus any change in storage (such as changes in soil moisture, groundwater, or surface water reservoirs)."

- *We have revised the statement to provide a more complete explanation of the water balance equation as recommended.*

8)   Page 18, Table 1: How do you interpret the potential reasons for why the performance metrics for equatorial, and warm temperate regions almost for all EXPs outperform those of other regions?

- *Thank you for your insightful question. The better performance metrics in equatorial and warm temperate regions can be attributed to their wetter conditions, as most models were developed for humid and semi-humid regions, which supports their suitability in this study (as mentioned in lines 468-470 of the preprint).*
  *Additionally, this may be due to effective land surface parameterization in these regions, which might not hold true for cold and arid areas. Lastly, the quality of ERA5-Land meteorological data may be limited in cold and arid regions, particularly in the absence of regional studies to confirm or refute this hypothesis.*

---

## Author Comment (AC2)

**Response to Anonymous Referee #2**:

The authors compare the impact of different runoff schemes on the hydrological simulations in different basins at a global scale. The paper is well-written and organized overall which is easy to read. The methodology is described in sufficient detail and provides a clear description of the results. I reviewed the paper, and I would highlight the following concerns.

Major comments:

1) Throughout the paper, it emphasizes the improvements in hydrological models, specifically highlighting this necessity. However, as the paper is written, it seems more focused on the evaluation of the different runoff generation schemes within the Noah-MP model. While this analysis is valuable, the paper could benefit from a more thorough exploration of how these findings can be applied to enhance global hydrological modeling, particularly in the context of discharge simulation, which I understand is the central aspect of the paper.

  - *Thank you for this insightful feedback. We agree that while the paper's focus on evaluating runoff generation schemes within the Noah-MP model is essential, the application of these findings to enhance global hydrological modelling and discharge simulation could be further clarified. Therefore, in response to the shortcoming of the original manuscript, we have added a dedicated section "3.5 Implications for Global Discharge Simulation" (given below) discussing how our findings could contribute to improve global hydrological models, emphasizing implications for discharge simulation and practical applications in water resource management and climate adaptation. We believe this addition strengthens the paper's relevance to global hydrological modelling advancements.*

*3.5 Implications for Global Discharge Simulation:*

[revised manuscript text omitted]

2)    Given that the results align with expectations across different regions and are not particularly surprising (lines 399 – 406, 461 – 462, and 469 – 470), it raises the question

of what the primary contribution of this study is. If the findings largely confirm well-established patterns, it would be helpful to clarify how this research advances current understanding or introduces novel insights into hydrological modeling. A clearer articulation of the contribution of this study would strengthen its impact and ensure that it is seen not just as a validation of existing knowledge, but as a meaningful step forward in hydrological research.

- *Thank you for your comment. We agree that the results were theoretically expected, and this study indeed confirms those expectations. However, as per the clarification on how this research advances hydrological modelling as a meaningful step forward, we have added a dedicated section "3.5 Implications for Global Discharge Simulation" to discuss the implications for hydrological simulation. See above.*

3) The paper offers a detailed analysis of the biases in different runoff generation schemes related to discharge, which is valuable. However, it would benefit from a deeper discussion of the underlying physical processes that contribute to these differences. By incorporating a more thorough exploration of the hydrological mechanisms driving these variations, the paper could provide a more comprehensive understanding of the findings and their implications.

- *Thank you for your insightful comment. In response, we have added a discussion (given below) that explores the underlying hydrological processes contributing to the biases observed in different runoff schemes. This section emphasizes how variations in the partitioning of precipitation into surface and subsurface runoff, as well as soil moisture dynamics, influence the distinct biases across schemes.*

*"The differences in biases between the runoff schemes are driven by how each scheme handles critical hydrological processes, particularly the partitioning of precipitation into surface and subsurface runoff, as well as the treatment of soil moisture dynamics. Although the formulas for some runoff schemes appear similar, the variations in specific parameters and their physical representations cause distinct biases to emerge.*

*For example, in the TOPMODEL-based schemes, while the surface runoff follows the same formula across both experiments, subsurface runoff differs in the way it is computed. In EXP1, subsurface runoff is influenced by the soil hydraulic conductivity of the first unsaturated layer above the water table, while in EXP2, the subsurface runoff is controlled by a fixed base flow coefficient and a fixed runoff decay factor. These differences lead to distinct sensitivities in how each scheme responds to variations in soil properties and terrain. The water table depth also plays a pivotal role in the calculation of subsurface flow in these schemes, introducing differences in regions with shallow or deep groundwater. These variations affect the magnitude and timing of runoff, which ultimately manifests as bias in the discharge simulation.*

*The surface runoff in the TOPMODEL-based schemes and BATS also shares the same formula; however, the differences lie in how the saturated area fraction (fsat) is calculated. In the TOPMODEL-based schemes, fsat depends on the water table depth and a runoff decay factor, which differs between the two schemes, while in BATS, it is computed as the fourth power of the degree of saturation in the top two meters of soil. This distinction between the two*

*approaches introduces variability in how surface runoff is generated across regions with differing soil moisture profiles and saturation conditions. The BATS scheme also handles subsurface runoff differently, using a free drainage approach where it is calculated as the product of soil hydraulic conductivity and (1 - fimp,max). This method leads to a distinct response to soil permeability and introduces varying biases depending on the frozen or compacted soil conditions.*

*For the other free-drainage schemes, including Schaake, VIC, XAJ, and Dynamic VIC, the calculation of subsurface runoff follows a similar approach. However, the key differences between these schemes lie in how surface runoff is generated. In the Schaake scheme, surface runoff is governed by an infiltration-excess mechanism, which depends on the total maximum holdable soil water content and the rate at which the soil can infiltrate water. This mechanism tends to produce lower biases in regions where infiltration-excess processes dominate, such as cold regions (Decharme, 2007).*

*The VIC scheme, on the other hand, calculates surface runoff based on the infiltration and maximum infiltration capacity of the soil, which introduces a different partitioning of rainfall into surface and subsurface components. The scheme's reliance on current soil moisture conditions, particularly in the tension water storage in the top layers of the soil, leads to varying biases depending on whether the region is experiencing wet or dry conditions. Similarly, XAJ introduces a unique approach by using a shape parameter to calculate surface runoff, which adjusts runoff generation based on the catchment's topographic and moisture characteristics. This leads to differences in performance depending on the terrain and hydrological profile of the region.*

*Dynamic VIC incorporates both infiltration-excess and saturation-excess runoff, further complicating the balance between surface and subsurface flow. The detailed soil moisture capacity parameters used in this scheme contribute to its dynamic nature but also make it more sensitive to inaccuracies in modelling infiltration and saturation, leading to large biases in discharge performance. The different ways each scheme handles these physical processes—whether through the treatment of soil moisture, the representation of surface and subsurface interactions, or the response to topographic and climatic variability—accounts for the differences in bias observed across the experiments. Understanding these physical distinctions is essential for improving the accuracy of runoff and discharge simulations, especially in regions with complex hydrological behaviour."*

Minor comments:

1)  it would be beneficial to replace informal phrases like: "On the flip side" (line 39) with more formal language.

    - *Thank you for your suggestion. We have revised the expression to a more formal tone as recommended ("On the other hand, …").*

2)  Section 2.1.1 would benefit from incorporating a table listing the different experiments, the runoff scheme used and their corresponding equations, making the text easier to read.

- *Thank you for your suggestion. We have incorporated a table (Table 1 below) in Section 2.1.1 listing the different experiments, the runoff schemes used, and their corresponding equations to enhance readability.*

*Table 1: Summary of the experiments (EXPs), runoff schemes, and corresponding equations*

| EXP | Runoff Scheme | Surface Runoff ($R_s$) Equation | | Subsurface Runoff ($R_{sub}$) Equation | |
|---|---|---|---|---|---|
| 1 | TOPMODEL with groundwater | $R_s = P_e \times \left[\left(1 - f_{imp}(1)\right) \times fsat + f_{imp}(1)\right]$ | (5) | $R_{sub} = (1 - f_{imp,max}) \times C_{baseflow} \times e^{-I_{topo}} \times e^{(-F_{decay} \times dwt)}$ | (6) |
| 2 | TOPMODEL with an equilibrium water table | Equation (5) | | Equation (6) | |
| 3 | Schaake | $R_s = P_e - Q_{infil,max}$ $\quad$ (7)
 $R_s = P_e \times \left[1 - \dfrac{w_{soil,tot} \times (1 - e^{-Kdt \times \Delta t})}{P_e \times \Delta t + w_{soil,tot} \times (1 - e^{-Kdt \times \Delta t})}\right]$ $\quad$ (8)
 The $Q_{infil,max}$ is further corrected for frozen soil as follows:
 $Q_{infil,max} = min(max(Q_{infil,max} \times f_{imp}; DK); P_e)$ $\quad$ (9) | | $R_{sub} = S_{drain} \times DK$ | (10) |
| 4 | BATS | Equation (5) | | $R_{sub} = (1 - f_{imp,max}) \times DK$ | (11) |
| 6 | Variable Infiltration Capacity (VIC) | If $i + P_e \geq i_{max}: R_s = P_e - W_{max} + W$ $\quad$ (12)
 If $i + P_e \leq i_{max}$:
 $\quad R_s = P_e - W_{max} + W + W_{max} \times \left[1 - \dfrac{i + P_e}{i_{max}}\right]^{(1+b)}$ $\quad$ (13)
 If $i_{max} = 0 : R_s = P_e$ $\quad$ (14) | | Equation (10) | |
| 7 | Xinanjiang (XAJ) | $R_s = (P_e \times A_{im}) + R \times \left(1 - \left(1 - \dfrac{S}{S_{max}}\right)^{E_x}\right)$ | (15) | Equation (10) | |
| 8 | Dynamic VIC | $R_s = R_{ie} + R_{se}$
 With:
 $R_{ie} = \begin{cases} if \ \dfrac{P - R_{se}}{f_m \times \Delta t} \leq 1, \\ P - R_{se} - f_{mm} \times \Delta t \times \left[1 - \left(1 - \dfrac{P - R_{se}}{f_m \times \Delta t}\right)^{b+1}\right] \\ otherwise, \\ P - R_{se} - f_{mm} \times \Delta t \end{cases}$ $\quad$ (16)
 $R_{se} = \begin{cases} if \ 0 \leq y < i_m - i_0, \\ y - \dfrac{i_m}{b+1} \times \left[\left(1 - \dfrac{i_0}{i_m}\right)^{b+1} - \left(1 - \dfrac{i_0 + y}{i_m}\right)^{b+1}\right] \\ if \ i_m - i_0 \leq y < P, \\ R_{se}|_{y=i_m-i_0} + y - (i_m - i_0) \end{cases}$ $\quad$ (17) | | Equation (10) | |

**$f_{imp}$(i): the ith soil layer impermeable fraction; $Q_{infil,max}$: the maximum soil infiltration rate; $w_{soil,tot}$: the sum of the maximum holdable soil water content in the unit of depth; Kdt: a coefficient for computing maximum soil infiltration rate; P: the amount of precipitation over a time step $\Delta t$; $f_{mm}$: the average potential infiltration rate over the 1-As area estimated based on the Philip infiltration scheme (Liang and Xie, 2003); $f_m$: the maximum potential infiltration rate; y: vertical depth; $i_0$: the point soil**

moisture capacity corresponding to the initial soil moisture; $i_m$: the maximum point soil moisture capacity; $C_{baseflow}$: a baseflow coefficient; $I_{topo}$: the gridcell mean topographic index.

3) In section 2.3, model evaluation, some paragraphs describe the actual models/data used (lines 243-248, and 272-276), I would suggest reorganizing these paragraphs in the corresponding sections.

   - *Thank you for your suggestion. We have reorganized the paragraphs as recommended: lines 243-248 have been moved to Section 2.1.2 (Input Data), and lines 272-276, which are associated with Table S1, have been moved to the supplement as Text S1.*

4) The text describing Figure 2 sometimes uses the "mm/year" units and sometimes %, I would suggest using the same units that appear in the figure (%) and indicate in brackets (mm/year), since using different units makes the text confusing.

   - *We have revised the text describing Figure 2 to consistently use "%" as in the figure, and included "(mm/year)" in brackets for clarity.*

5) Figure 3 could be improved to show only the basins mentioned in the text, this would make it easier to follow the findings described in the text directly in the figure. The rest of the basins could be included as a complementary figure.

   - *Thank you for your feedback. We have revised Figure 3 to include only the basins mentioned in the text for better clarity (figure below), while moving the original figure to the supplement as Figure S1.*

[Figure]

*Figure 3: Mean seasonal cycle of runoff (mm) and river discharge (m3/s) simulated by the different Noah-MP runoff schemes and CaMa-Flood, for 6 selected river basins representing four climate regions (cold, warm temperate, equatorial and arid). Discharge data includes simulated and observed values (obs) for the period 1985–2023. Observation years contributing to the monthly mean vary depending on their availability, with a minimum of 5 years per catchment.*

6)     Lines 357 – 362, While the central idea is clear, the text is somewhat difficult to follow due to its current structure and phrasing. I recommend restructuring this paragraph to improve coherence and enhance the link with the next paragraph.

-       ***Thank you for your helpful feedback. We have restructured the paragraph in lines 357-362 to improve coherence and enhance the link with the following paragraph, as follows:***
        ***"Across many basins, the seasonal cycles of runoff and discharge generally agree. However, a noticeable lag often exists between the peak runoff and peak discharge, especially in large river basins like the Amazon. This lag, which can extend up to three months (Liang et al., 2020; Sorribas et al., 2020), is due to the natural routing process within the river network. This process involves the time it takes for water to travel through the system and the storage effects within river channels, depending on basin characteristics such as size, shape, drainage density, river length, and slope. In some cases, this lag could also reflect limitations in the CaMa-Flood routing model, particularly for large-scale river basins where routing dynamics are complex. A detailed sensitivity analysis of the routing parameterisation (such as river velocity, roughness coefficients, or floodplain dynamics) could offer valuable insights into how model-specific limitations impact the timing of peak discharge. This could be an important direction for future research, with the potential to enhance model performance in accurately simulating discharge timing."***

7)     Line 405: David et al., 2019. That reference was previously cited in the text and should be properly referred to again in this section. Instead of using a link, it would be more appropriate to use the established citation format.

-       ***We have revised this line to properly reference David et al. (2019) using the established citation format.***

8)     I suggest reordering Section 3.4 to enhance clarity. Starting with the description of the global performance metrics (and not at the end as it is presented now) and then moving to the findings in detail for each of the regions (cold, warm, etc.), would improve section structure.

-       ***Thank you for your suggestion. We have reordered Section 3.4 to start with the description of the global performance metrics, followed by detailed findings for each region.***

9)     The authors highlight that further improvements are necessary, such as refinement across diverse climatic regions, and calibration at finer resolutions (lines 336-338, 490 – 494). However, they could provide suggestions or hypotheses for improving global hydrological models and discuss potential refinements in more detail.

-       ***Thank you for the suggestion. We agree with the need for detailed refinement strategies and have provided examples (see below) on potential refinements to improve global hydrological models, including calibration and region-specific adjustments.***

*"An area for improvement would involve calibrating these parameters, particularly at finer resolutions, to more precisely simulate runoff behaviour across diverse regions. For example, fsatmax is often set as a global mean, but recent studies, such as (Zhang et al., 2022), illustrate that using spatially variable values informed by remote sensing data (e.g., GIEMS-2) could yield more accurate regional simulations. Similarly, the fixed soil depth used in each scheme could be improved through further spatially variable parametrization within Noah-MP, which may help modulate runoff according to regional soil profiles and enhance the model's representation of subsurface flow dynamics. By exploring such refinements, future applications of these models could achieve better performance, especially when simulating high-flow events critical for flood risk assessments and water resource management."*

10) While the conclusions offer valuable insights into the regional outcomes, as they are written, it seems that they do not adequately reflect the global perspective outlined in the paper: "Our study transcends the boundaries of individual schemes and specific regions, highlighting the need for a holistic assessment…". It would be beneficial to rephrase the conclusions to present a more unified global argument, aligning with the objective of the research

- *Thank you for this observation. We have revised the conclusions to present a more unified global perspective:*

*4 Conclusions*

[revised manuscript text omitted]